# Development of a Culinary Medicine Toolkit to Improve Implementation of Virtual Cooking Classes for Low-Income Adults with Type 2 Diabetes

**DOI:** 10.3390/healthcare12030343

**Published:** 2024-01-30

**Authors:** David Ai, Natalia I. Heredia, Vanessa Cruz, Diana C. Guevara, Shreela V. Sharma, Dolores Woods, Melisa Danho, John Wesley McWhorter

**Affiliations:** 1School of Medicine, Baylor College of Medicine, 1 Baylor Plaza, Houston, TX 77030, USA; david.ai@bcm.edu; 2Department of Health Promotion Behavioral Sciences, University of Texas Health Science Center at Houston School of Public Health, 7000 Fannin, Houston, TX 77030, USA; 3Whole Foods Market Foundation, 550 Bowie St., Austin, TX 78703, USA; 4The Michael & Susan Dell Center for Healthy Living, University of Texas Health Science Center at Houston School of Public Health, 1200 Pressler St., Houston, TX 77030, USA; diana.guevara@uth.tmc.edu (D.C.G.);; 5Department of Epidemiology, University of Texas Health Science Center at Houston School of Public Health, 1200 Pressler St., Houston, TX 77030, USA; shreela.v.sharma@uth.tmc.edu; 6Center for Health Equity, University of Texas Health Science Center at Houston School of Public Health, 1200 Pressler St., Houston, TX 77030, USA; 7Suvida Healthcare, 515 Post Oak Blvd, Suite 510, Houston, TX 77027, USA; wmcwhorter@suvidahealthcare.com

**Keywords:** culinary medicine, virtual, toolkit, diabetes, nutrition, patient education, education

## Abstract

Culinary medicine (CM) addresses diseases through nutrition and culinary education. To promote access to educational material for people with diabetes and engagement in virtual classes, we created a virtual culinary medicine toolkit (VCMT) sensitive to literacy levels and language preferences. The VCMT was developed to accompany existing virtual CM programs and help improve participant interaction and retention, offering educational materials for providers and participants. The provider VCMT offers level-setting education to reduce mixed nutrition messaging, including educational resources discussing inclusive nutrition and mindful eating topics. Each handout has a QR code and link to engaging, animated videos that provide further explanation. The participant VCMT offers a range of fundamental cooking skill videos and infographics, including knife skills and preparing whole grains and healthy beverages. Participant handouts and animated videos, which are played during the virtual CM class, allow participants to learn more about diabetes management and food literacy topics, including interpreting nutrition labels, and are employed during a CM to facilitate discussion and reflection. The animated videos replace a traditional slide-based lecture, allowing space for patient-centered facilitated discussions during virtual cooking sessions. The VCMT could guide the development of virtual CM interventions to shift learning from lecture-based to patient-centered discussions via a visual and inclusive medium.

## 1. Introduction

Over 37 million (11.3%) adults in the United States have type 2 diabetes mellitus (T2DM), of which a suboptimal diet is a major risk factor [1,2]. Furthermore, the prevalence of DM is forecasted to increase by more than 50% by the end of this decade [3]. Moreover, low health literacy is associated with poorer glycemic control and general diabetes knowledge in patients with diabetes [4,5,6,7]. Patients may recall or understand as little as half of the information provided by physicians [5], a concern exacerbated by the fact that those with T2DM often have low levels of health literacy [8]. The increasing prevalence of T2DM and low health literacy levels warrant an urgent need for improved, accessible nutrition education [9,10].

Culinary medicine (CM) is an emerging educational and nutritional approach that adds to current approaches and interventions by incorporating both practical hands-on preparation skills and the scientific knowledge of how nutrition and dietary patterns affect health [11,12]. CM provides a structured approach to develop patients’ perception of food and cooking as a component of their healthcare [11]. CM interventions have demonstrated a significant pre-to-posttest reduction in hemoglobin A1c levels, diastolic blood pressure, and cholesterol levels in people with T2DM [13,14]. However, there still exists a need to include more culturally relevant topics in the design and implementation of CM interventions for minority populations [15], which the toolkit described in this study seeks to address.

In the current model of CM interventions, there is a missed opportunity to further engage with participants and facilitate discussion during cooking downtime. Thus, while other online diabetes education toolkits exist [16], to our knowledge, there are no existing supplementary online materials designed to complement CM intervention sessions. Specifically, our virtual culinary medicine toolkit (VCMT) materials are designed to reinforce skills taught during the CM session. For example, demonstrations of knife skills at the live CM sessions are reinforced with knife skills videos provided in the VCMT, a resource that participants can use multiple times as a reference. Learning through observation and self-controlled practice is extremely effective for acquiring and reinforcing new skills [17]. Moreover, given the context of COVID-19 and the shift to virtual delivery for many behavioral interventions, this study describes the development of online supplementary materials for both patients and providers, known as the VCMT, for use alongside CM interventions for low-income patients with T2DM.

## 2. Materials and Methods

The VCMT was developed to provide easily accessible diabetes and nutrition education materials for participants and providers and to improve participant interaction, engagement, and retention in CM interventions. The main CM curriculum follows the USDA MyPlate nutrition guidelines and is informed by social cognitive theory (SCT) constructs, including outcome expectations, self-efficacy, behavioral capability (knowledge & skills), self-regulation, and observational learning. These constructs were previously used to guide A Prescription for Healthy Living (APHL), a clinic-led food prescription program and culinary medicine intervention for patients with T2DM [13]. To develop the program, we used intervention mapping to systematically map curriculum constructs to the desired behavioral, environmental, physiological, and psychosocial outcomes of the APHL intervention [13]. This program served as a theoretical foundation for comprehending the elements that facilitate and sustain changes in behavior for this VCMT [18]. These include expectations regarding the taste of nutritious foods, knowledge about healthy eating, self-efficacy in preparing healthy foods, proficiency in cooking nutritious meals, perceived social encouragement for adopting healthy dietary habits, and normative beliefs related to consuming nutritious foods (see Table 1) [19]. This VCMT builds on and adapts the print materials and handouts from the APHL program to a virtual format.

The content and design of the toolkit videos were determined by a multidisciplinary team of experts, including registered dietitians (RDs), chef-RDs, behavioral scientists, and public health professionals, who drew upon their experience from conducting previous CM interventions and from working within the Nourish program, an adjunct training program for dietetic interns at the University of Texas Health Science Center at the Houston School of Public Health consisting of a holistic garden, a culinary research and demonstration kitchen, and a patient simulation lab [13,20]. The process of content creation for each video began with a team member drafting a list of general objectives (see Table 2 and Table 3) based on the current literature. Next, an outline was created detailing the video’s main points and a storyboard. A video script was then drafted and created using Vyond (utilized in 2020 and 2021), a video animation software. Each video underwent three rounds of review by the entire team, during which the videos for animatics, dialogue, music, and sound effects were examined. After these three rounds, the multidisciplinary team reviewed the videos once again to finalize, ensuring that the content and delivery were easy to comprehend, and assessing flow, congruency of animations with the script, and overall appearance [13,20]. 

Importantly, the team prioritized diversity and inclusion during video development, creating characters that represent various genders, sexes, races, and ethnicities to ensure broad representation. Voiceovers were performed by individuals of diverse gender and cultural identities, further emphasizing inclusivity and cultural congruence. The videos were translated from English to Spanish by native Spanish speakers, with an additional round of review to ensure accuracy and cultural sensitivity [21]. The result is a comprehensive and inclusive educational resource that aligns with the principles of effective and engaging learning experiences.

The handouts in the VCMT were created based on the video scripts and contained both information and activities for participants to complete after each CM session. After the team members individually drafted sections of the handouts, each section went through five rounds of review by the entire team. A third-party graphic designer with a background in public health was contracted to refine the visual content of the handouts. The Spanish-language content of the VCMT (see Figure 1 and Figure 2) was created with the input of several native Spanish speakers with expertise in the subject and sought to convey the content effectively as opposed to simply being direct translations from the English version. Unlike traditional videos, the VCMT offers a dynamic learning experience by incorporating handouts, worksheets, and culinary skill walkthroughs that pair directly with the live CM sessions. By embracing multimodal learning, the VCMT aims to improve knowledge retention and learning satisfaction [22,23].

The VCMT was designed to complement a virtual, synchronous CM intervention that was adapted from an in-person CM curriculum [13,20]. We are pilot testing this VCMT curriculum alongside a virtually delivered CM intervention (Nourishing the Community through Culinary Medicine), adapted during COVID-19 from an in-person curriculum [13,20]. The VCMT is made available to participants ahead of time, providing online access to educational materials, including all handouts and videos. VCMT materials were distributed to participants via QR codes, a website, and a link through email, promoting accessibility through both computers and mobile devices. A print version of the VCMT handouts with QR codes for the videos was mailed to the participants’ homes, and it included all sessions’ recipes and shopping lists, along with handouts that participants could fill out on their own time. As it was available to the participants beforehand, the VCMT served as a reference tool that could be used during and after the sessions. During each of the five virtual cooking sessions that are part of Nourishing the Community through Culinary Medicine, particularly while there is downtime and participants wait for a certain ingredient or a step to be cooked/completed, the instructor plays select videos, followed by a facilitated discussion of the video content. These animated videos serve to replace traditional slide-based lectures and facilitate more engagement in the virtual setting. As part of this pilot testing, we are collecting qualitative post-intervention interviews with selected participants to understand their impressions of the VCMT.

The online version of the VCMT is open access and without a paywall, providing a free touchpoint for participants to review content or skills taught during synchronous, virtually delivered CM intervention sessions and for instructors to have access to standardized content and resources. The VCMT can be found on the Nourish program website (https://sph.uth.edu/research/centers/dell/nourish/ (accessed on 26 November 2023)), and videos from the VCMT can also be found on The Michael and Susan Dell Center for Healthy Living at the University of Texas School of Public Health YouTube Channel (https://youtube.com/@msdcenter (accessed on 26 November 2023)).

## 3. Results

The patient’s VCMT contains videos and a master handout in English and Spanish. Video topics include an overview of carbohydrates, tips for grocery shopping, knife skills, and eating balanced meals (see Table 2). The patient handout is structured based on the order of the five CM sessions (see Table 3), with informational material incorporated throughout the handout. Patients can use this handout to prepare for each CM session and as a reference in the future.

The provider VCMT contains a master handout (see Figure 3 and Table 3) and videos in English. The provider VCMT addresses topics for successful provider–patient communication on culturally sensitive, holistic discussions on dietary behaviors. Video topics include an overview of the social determinants of health, providing culturally sensitive nutrition counseling, and the role of registered dietitian nutritionists (RDNs) on the healthcare team (see Table 2). The handout contains tips on how to facilitate sensitive behavior change discussions with patients and presents information designed to encourage providers to consider the external factors behind patients’ dietary choices (see Figure 4).

Participants who have already completed the Nourishing the Community through Culinary Medicine trial were asked about their impressions and satisfaction with the online materials, including their access to the virtual toolkit and the animated videos. Most participants did not encounter difficulties accessing the online materials, with one individual mentioning: *“It’s so easy for me to find all that information. It’s because they sent it directly to me by email”.* Similarly, another individual said: *“I found it easy. They just give you all the information and you just click on it and you go straight to the access”.* Participants appeared to like the toolkit, with one saying: *“the toolkit was wonderful, the videos, all of it was super easy and also accessible. I enjoyed that a lot”.*

## 4. Discussion

CM interventions have the potential to address the behavioral and psychosocial aspects of chronic conditions like T2DM, including dietary habits, attitudes toward healthy eating, enhanced culinary skills, and increased confidence in cooking and maintaining a healthy diet [28,29,30,31,32]. The systematically mapped SCT-based curriculum constructs and patient outcomes provided a theory-based framework for the VCMT, particularly in initiating and maintaining behavioral, environmental, physiological, and psychosocial outcomes [13]. Applying such an SCT-based approach allowed us to incorporate both behavioral theory and evidence-backed knowledge in crafting the VCMT, helping to better leverage the factors influencing behavior change and aiming to establish enduring and sustainable transformations [18].

The development of the VCMT was driven by the need for an innovative approach to patient education and engagement in the realm of T2DM management and food literacy. The patient VCMT addresses this gap by offering a patient-centered approach that extends beyond the duration of the CM intervention, aiming to maintain participant engagement and enhance learning outcomes. The toolkit employs a multimodal learning strategy by incorporating open access videos, handouts, and recipes to accommodate individuals’ diverse learning preferences. This approach ensures that participants can access information before, during, and after CM interventions, fostering continuous learning and reinforcement.

Importantly, the VCMT serves as a guide for the development of future virtual CM resources for interventions, emphasizing a shift from lecture-based learning to patient-centered discussions through a visual and inclusive medium. By leveraging animated videos, handouts, and recipes, the VCMT not only imparts crucial information but also creates a dynamic and engaging learning environment, fostering a deeper understanding and retention of knowledge. This toolkit also aims to be sensitive to patients’ literacy levels and language preferences. We prioritized ensuring that the educational materials are accessible and understandable for individuals with varying literacy levels and different language preferences. Our future work also involves adapting the VCMT to other languages.

The provider sections of the VCMT aim to offer more standardized, level-setting nutrition and CM education, encompassing cultural sensitivity, communication strategies, and holistic approaches to patient care. This emphasis on communication skills, empathy, and patient-led communication reflects an awareness of the evolving dynamics in healthcare interactions and the importance of a patient-centered approach to foster effective communication between providers and patients for more meaningful and sustainable health outcomes [33]. The provider sections are particularly useful considering the emerging importance of CM and the shortage of healthcare professionals educated in this field; we hope they serve as a pioneering resource designed to broaden healthcare providers’ knowledge base and fill a critical gap in healthcare education [22]. In sum, this VCMT contributes towards the goal of an informed healthcare workforce, with providers that are well-equipped to promote dietary habits conducive to overall health and wellness.

The VCMT stands at the forefront of patient-centered, virtual culinary education, providing a comprehensive and accessible resource for individuals managing diabetes. Its emphasis on multimodal learning, open access information, and the promotion of sustainable, family-oriented healthy eating behaviors positions it as a valuable tool for improving long-term health outcomes and empowering patients in their journey towards better dietary patterns [22,23]. However, there are some limitations in the use of the VCMT. Video streaming requires stable, consistent access to technology and the internet, which not all people participating in CM interventions may have. There are also potential limitations in integrating this VCMT with pre-existing CM interventions that may follow different curricula, have different target populations, or target different chronic diseases.

The VCMT is being tested as part of a full-scale CM trial, which will quantitatively assess biometric, psychosocial and behavioral outcomes. Qualitative feedback will additionally explore perceptions of accessibility, animated videos, online resources, feasibility, VCMT recipes, community, and the cultural diversity of dishes taught. Thus, we will aim to further assess the impact of multimodal learning on participant engagement, learning outcomes, and effectiveness outcomes. 

## 5. Conclusions

The VCMT contains information in both English and Spanish for patients and providers on a wide range of information for patients with T2DM. The VCMT was designed to facilitate the transition to online CM sessions after the COVID-19 pandemic, providing easily accessible online information. Our VCMT is innovative in that it takes advantage of downtime during live cooking sessions and provides patients with open access to multimodal educational materials to reinforce knowledge and skills acquired during synchronously delivered sessions.

Although it was mainly designed for low-income Hispanic adults in Texas with T2DM, the VCMT was developed considering diversity, inclusion, and cultural sensitivity and could potentially be used with other populations. In the future, we would like to gather feedback from community members to drive further modify the VCMT. Moreover, the processes described in this study can be used to adapt the VCMT to other target populations. The VCMT will soon be used by partner sites as part of a biweekly food prescription program, independent of a culinary medicine program, showcasing the potential of this toolkit to be used in a variety of ways moving forward.

## Figures and Tables

**Figure 1 healthcare-12-00343-f001:**
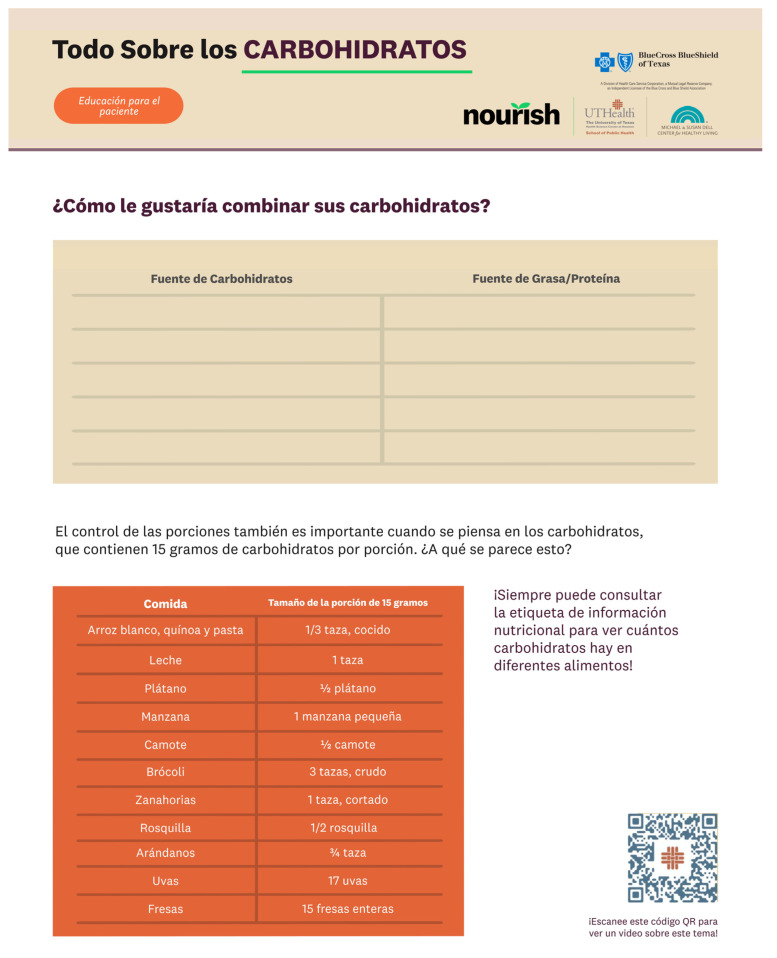
An excerpt from the Spanish version of the patient VCMT.

**Figure 2 healthcare-12-00343-f002:**
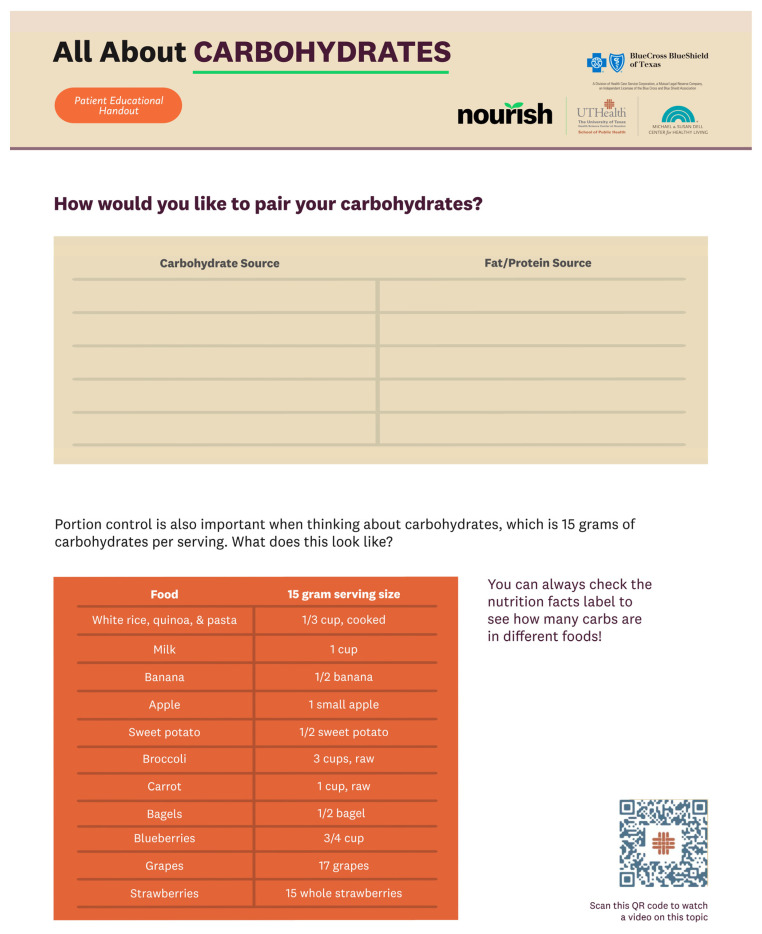
The English version of Figure 1 for reference.

**Figure 3 healthcare-12-00343-f003:**
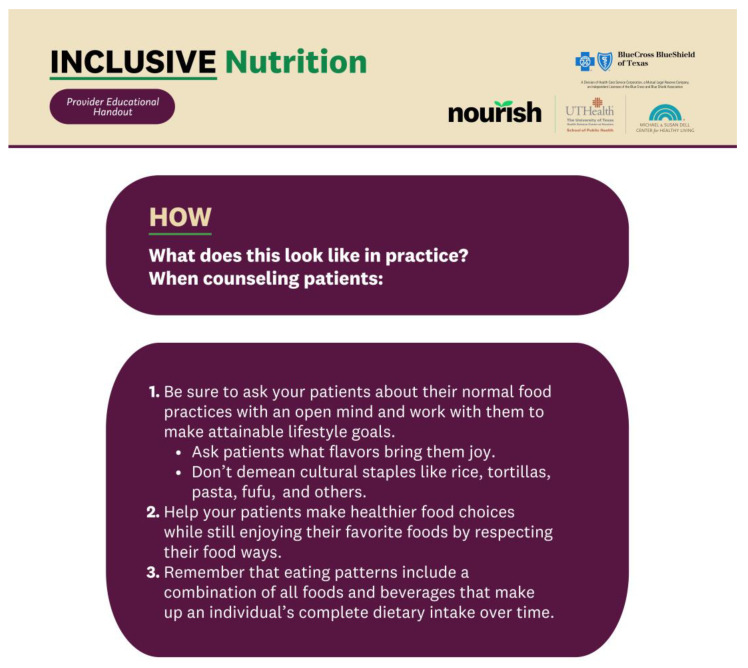
The provider VCMT outlines tips to facilitate communication with patients on many topics, including culturally sensitive nutrition, encouraging exercise, and delivering patient-centered care.

**Figure 4 healthcare-12-00343-f004:**
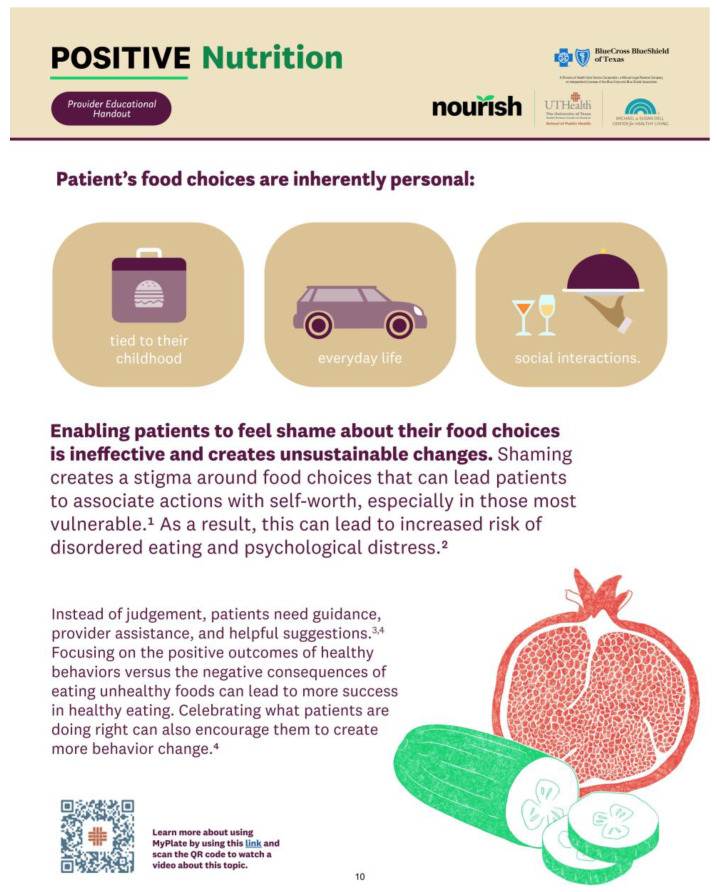
The provider VCMT serves to encourage providers to consider external factors that may affect patients’ dietary patterns. Please note that superscripts 1–4 in the image correspond citations [24,25,26,27].

**Table 1 healthcare-12-00343-t001:** SCT applied to the design of the VCMT.

Constructs Applied to VCMT Development
Outcome expectations of the taste of healthy foods Knowledge of healthy eating Skills for preparing healthy foods Self-efficacy for preparing healthy foods Perceived social support for consuming healthy foods Normative beliefs around eating healthy foods

**Table 2 healthcare-12-00343-t002:** VCMT video content.

**Patient Video Topics**	**Description**
MyPlate is your plate	Describes what MyPlate is and its healthy food components.
All about carbohydrates	Provides an overview of how carbohydrates fit into a healthy dietary pattern.
What is diabetes?	Explains diabetes diagnoses and what to expect for dietary changes.
Blood sugar overview	Explains how what you eat affects your blood sugars and how to maintain equilibrium.
Controlling low blood sugar	Provides tips for managing diet and behaviors to prevent or remedy low blood sugar.
Controlling high blood sugar	Provides tips for managing diet and behaviors to prevent or remedy high blood sugar.
Why beverages matter	Explains how sugar in beverages adds up.
Diabetes and medications	Provides an overview of why patients should take their medication and how food might interact.
Meeting with a registered dietitian	Provides an overview of what it is like to meet with an outpatient dietitian.
Mindful eating and snacking	Provides patients with the understanding of why they are eating or want to eat and to be mindful and enjoy the food they consume.
It is a family affair	Provides patients with the tools to help make changes to their dietary patterns as a family.
Picky kids	Describes how patients can avoid making separate meals for kids while encouraging healthy eating for all.
Beyond the scale	Encourages patients to focus on sustainable, long-term healthy eating behaviors as opposed to the number on the scale.
Goal setting	Introduce patients to the idea of making small behavioral change goals and increasing them gradually.
Grocery lists	Teach patients how to create a grocery list; give tips on what kind of products to purchase for each food category.
Nutrition facts labels	Teach patients how to read nutrition facts labels.
Stocking a healthy pantry	Provide an overview of common pantry staples for each food category.
**Provider Video Topics**	**Description**
Inclusive nutrition messaging	Discusses how healthy food can align with all cultures’ foods.
Social determinants of health	Provides an overview of food insecurity and its association with chronic disease
Patient–provider communication	Provides an overview of how to communicate with patients utilizing empathy and patient-led communication.
Positive nutrition	Discusses the provider’s role in providing consistent and positive nutrition messaging to patients.
Dietitian on the team	Explains what an RD does and why they are a valuable part of the patient care team.
Mindful eating	Provides an overview of mindfulness in relation to healthy food behaviors.
Physical activity	Discusses how providers can encourage movement in conjunction with healthy eating.
Beyond the scale	Describes external factors to consider when working with obese patients and emphasizes health factors other than weight that contribute to health status.
Flavors, not formulas	Provides suggestions for providing positive, personalized nutrition recommendations.
**Patient Video Topics**	**Description**
MyPlate is your plate	Describes what MyPlate is and healthy food components.
All about carbohydrates	Provides an overview of how carbohydrates fit into a healthy dietary pattern.
What is diabetes?	Explains diabetes diagnoses and what to expect for dietary changes.
Blood sugar overview	Explains how what you eat affects your blood sugars and how to maintain equilibrium.
Controlling low blood sugar	Provides tips for managing diet and behaviors to prevent or remedy low blood sugar.
Controlling high blood sugar	Provides tips for managing diet and behaviors to prevent or remedy high blood sugar.
Why beverages matter	Explains how sugar in beverages adds up.
Diabetes and medications	Provides an overview of why patients should take their medication and how food might interact.

**Table 3 healthcare-12-00343-t003:** VCMT handout content.

**Patient Sections**	**Subsections**	**Recipes**
Session 1	Healthy eating, goal setting	Roasted chicken thighs and vegetables with quinoa; shopping list.
Session 2	All about carbs, diabetes management	Barley pilaf; Greek style baked white fish and vegetables; shopping list.
Session 3	Healthy habits for the family	Green turkey chili, coleslaw with chili lime dressing; shopping list.
Session 4	Planning healthy meals	Orange chicken and vegetable stir-fry; shopping list.
Session 5	Beyond the scale, mindful eating, meeting with an RDN	Ground beef and pasta skillet; garlic and herb butter broccoli; shopping list.
Cooking Skills and Tips		
Animated Videos and Resources		
**Provider Sections**	**Description**	
Beyond the scale	Discusses talking about weight with patients.	
Flavors, not formulas	Explains the need to reframe the way we talk about eating healthy by talking about the flavor and not formulaic diets.	
Mindful eating	Emphasizes that focusing on what you eat can make a difference by acknowledging fullness cues, controlling portion sizes, and stopping mindless eating.	
Inclusive nutrition	Emphasizes that implementing inclusive nutrition begins with cultural humility and understanding that patients are experts in what they eat and why.	
Positive nutrition	Focuses on the positive outcomes of healthy behaviors that can lead to more success in healthy eating.	
Physical activity	Emphasizes that physical activity is a modifiable risk factor that can reduce mortality of individuals by up to 30%.	
Patient communication	Teaches five components to incorporate in practice to create patient-centered care and open communication.	
Registered dietitians on the team	Explains that registered dietitians can work on interprofessional teams in a multitude of healthcare settings to deliver a diversified care plan for patients.	
Cooking skills	Teaches some healthy cooking skills and techniques.	

## Data Availability

The toolkit is available at https://sph.uth.edu/research/centers/dell/nourish/research-resources/toolkit (accessed on 26 November 2023).

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
