# Peer review of "Development of a Culinary Medicine Toolkit to Improve Implementation of Virtual Cooking Classes for Low-Income Adults with Type 2 Diabetes"

_healthcare, 2024, doi:10.3390/healthcare12030343_

Round 1

Reviewer 1 Report

Comments and Suggestions for Authors

In “Development of a culinary medicine toolkit to improve implementation of virtual cooking classes for low-income adults with uncontrolled type 2 diabetes”, the authors describe an interesting and potentially beneficial open-access tool that they have developed for nutrition education aimed at T2DM patients.

However, to my advice the report does not meet the standards for publication in scientific literature, even as a technical note, due to the complete lack of any outcome measure about either feasibility or benefits related to implementation of the described tools.

The authors state the toolkit they have developed improves existing virtual CM programs, but they cannot provide any evidence to that: how do we know that this is indeed the case, if we don’t have any impact measurements? For all we know, patients with T2DM could gain no benefit whatsoever after using this toolkit, or even be worse off! 

Further, the authors advocate for the need or more fine-tuned educational tools that, for example, account for “different literacy levels”, “reduce mixed nutrition messaging”, or “include more culturally relevant topics for minority populations”. But again, they provide no evidence that the tool they have developed actually makes any difference or improvement in such areas.

In my opinion, the authors should at least test their toolkit on a small scale, with a group of T2DM patients, and report the results of - at least - a satisfaction questionnaire and some feasibility outcomes monitored during implementation. Even better, they should perform a knowledge and skills questionnaire before and after the program, to see if there was indeed any improvement which confirms that the toolkit is useful and has some differential advantages compared to already existing education tools. 

Otherwise, just wait for the results of the full scale trial that is mentioned in the conclusion section. At the present state, the paper just looks to me as an advertising flyer for the program.

Comments on the Quality of English Language

No comments. 

Author Response

The primary objective of our current paper is to present and discuss the development of the Virtual Culinary Medicine Toolkit (VCMT). We acknowledge the absence of specific outcome measures related to feasibility or benefits in the manuscript. However, we would like to emphasize that the primary focus of this paper is on the theoretical framework, implementation strategies, and content design of the VCMT. As you rightly pointed out, our paper emphasizes the need for more fine-tuned educational tools, especially those accommodating different literacy levels, reducing mixed nutrition messaging, and incorporating culturally relevant topics for minority populations. While we agree that outcome measures are crucial and appreciate your suggestions for testing the toolkit on a small scale, we want to clarify that our intention in this paper is to lay the groundwork for the VCMT and its theoretical underpinnings.  Given that the VCMT is a new method of supplementing CM interventions, to our knowledge, there was little existing data to rely on as we created the VCMT. However, it is important to note that the VCMT was created based on a pre-COVID culinary medicine intervention that has already been tested and has favorable patient outcomes (citation can be found in the updated draft). Since we are focusing on the creation, design, and implementation of the VCMT, we have expanded on the theoretical framework of the VCMT (Methods section).

We acknowledge the importance of empirical validation and intend to conduct a comprehensive evaluation, including satisfaction questionnaires, feasibility assessments, and knowledge and skills questionnaires, in future studies. Your recommendation aligns with our long-term plan, and we thank you for highlighting these crucial considerations.

Please note that this manuscript has been updated to specify that “the VCMT is being tested as part of a full-scale CM trial, which will quantitatively assess biometric outcomes and qualitatively assess participants’ feedback on the VCMT. Specifically, feedback will include interview questions about accessibility, animated videos, online resources, feasibility, VCMT recipes, community, and the cultural diversity of dishes taught.”

Furthermore, we have also gone ahead and included a few quotes obtained from participants who already completed the trial to provide insight into participants’ experience with the VCMT (lines 170-178).

Reviewer 2 Report

Comments and Suggestions for Authors

This article nicely highlights the innovation of VCMT in that it is patient-centered and provides open, multimodal educational materials to reinforce skills taught at the CM session. But there are still a few problems to be solved.

1. Define the VCMT acronym in 63 lines, should be changed to it for the first time in the position of the line (56) write full name after the abbreviations.

2. The VCMT to pilot test, is expected in the crowd can get the result?

3. The contrast between the advantages of VCMT and other interactive educational videos can be expressed more clearly.

4. The main topic of the article is to intervene in patients with type 2 diabetes by improving the cooking skills of low-income patients. What are some of the features of this toolkit that are specific to low-income people? Does it work?

5. The toolkit is aimed at the patient's portrait?

6. The letter spacing is not consistent in Figure1-3 lead to less beautiful.

Author Response

  1. Full name now written in line 57
  2. The VCMT has been pilot tested, and results will be published in a separate publication detailing the CM intervention and implementation of the VCMT.  We have added some quotes from participants regarding their experience with the VCMT in lines 170-178. However, in this technical note, we focus on the development of an innovative toolkit for CM interventions.
  3. We have emphasized the fact that the VCMT offers more than educational videos, including worksheets and handouts. Unlike many videos, the VCMT is also open access and without a paywall (also stated in Materials and Methods). Furthermore, the VCMT is also stated to have information and activities for participants to complete after each CM session. By emphasizing the multimodal learning that the VCMT supports, we aim to improve knowledge retention and learning satisfaction more than interactive educational videos alone.
  4. The online, open-access played a large part in the VCMT’s accessibility. The translation into Spanish was also an important undertaking for our target population. In the Discussion section, we do acknowledge that not all CM participants may have internet access. As mentioned in the paper, we have taken the step to mail print versions of the VCMT handouts for our CM intervention participants.
  5. This toolkit is aimed at both patients and providers (see end of Introduction)
  6. We are unsure of what this suggestion is for; the figures are screenshots of the already disseminated and published toolkit. Are you referring to the text alignment of the captions?

Reviewer 3 Report

Comments and Suggestions for Authors

This is an interesting manuscript describing innovative and culturally sensitive methods to introduce a virtual culinary medicine toolkit to low-income adults with type-2 diabetes. The manuscript is well-written, and the contents of the virtual platform are well-designed. I only have a few minor suggestions to improve the manuscript before acceptance.

1)      In the introduction (line 2), the authors write that type-2 diabetes is caused by a suboptimal diet. Type-2 diabetes is a result of either less insulin production by the pancreas or the development of insulin resistance in liver, fat, and muscle cells.   Although a suboptimal diet can lead to an early onset of the disease and exacerbate type-2 diabetes-related negative symptoms, it is not causative.  I recommend revising the sentence.

2)      The manuscript lacks quantitative data to demonstrate the effectiveness of the virtual platform, which in my opinion is a key weakness of this study.  I suggest that the authors should try to incorporate surveys or a quiz to test if participants were better informed after using the virtual platform and experienced an overall benefit. This data can then be analyzed to assess the benefits and drawbacks of the virtual toolkit.

3)      In author contribution section states that some authors contributed to data analysis. This statement is counterintuitive as the manuscript does not include any quantitative data. I recommend revising the statement.

4)      In the manuscript, the authors mention poor literacy as the reason for suboptimal nutritional practices in low-income communities. However, food inaccessibility and unaffordability are critical factors contributing to malnourishment. The authors may include information on community food banks or pantries, where low-income families can access affordable food.

5)      Incorrect insertion of citations on lines 36 and 60.

6)      Missing punctuations on lines 70 and 93.

Comments on the Quality of English Language

Minor editing is recommended, with attention to proper punctuation, citations, and tense agreement.  

Author Response

  1. We have revised the statement in the Introduction to state that a suboptimal diet is a major risk factor for T2DM.

  2. We have gone ahead and added quotes from participants who have already completed their sessions of the culinary medicine intervention regarding their experience using the VCMT (lines 170-178). We have also emphasized that the VCMT is being tested as part of a full-scale CM trial, which will quantitatively assess biometric outcomes and qualitatively assess participants’ feedback on the VCMT (line 227).
  3. Statements removed.
  4. We are receptive to suggestions for improving our VCMT; this paper focuses on the development of the VCMT and its content, and we thank you for your suggestion to increase the breadth of our VCMT content. This is something we will consider implementing in future versions of the VCMT.
  5. Addressed
  6. Addressed

Round 2

Reviewer 1 Report

Comments and Suggestions for Authors

I appreciate the effort made by the authors to improve upon the context and perspective of the manuscript.

While I still see no particular interest in this manuscript until specific outcomes are available, publication at this point is an editorial choice that is not up to me to make but to the journal editors. From a strictly content-wise perspective, I see no problems with the manuscript in its present form.

However, a program for nutrition education that teaches what to eat, might appear to have a potential conflict of interest with a company that sells food (especially since each of the learning modules of the program explicitly includes a “Walmart Shopping List”, and that some recommendations could be considered controversial, such as the statement, in session 3, that “Diet drinks can be good substitutes for sugar sweetened beverages”), and a program aimed at reducing a disease, with a healthcare provider whose clients are sick patients. This is not a problem for publication, but I strongly recommend that a mention be made in the Conflict of Interest statement at the end of the manuscript, that two of the authors belong to such private companies.

Author Response

The manuscript itself has no mention of Walmart or substitution of sugar sweetened beverages with diet beverages, and the conflict of interest statement has been updated.